# Controlling Covariate Shift using Equilibrium Normalization of Weights

## Abstract

We introduce a new normalization technique that exhibits the fast convergence properties of batch normalization using a transformation of layer weights instead of layer outputs. The proposed technique keeps the contribution of positive and negative weights to the layer output in equilibrium. We validate our method on a set of standard benchmarks including CIFAR-10/100, SVHN and ILSVRC 2012 ImageNet.

## 1 Introduction

The introduction of normalizing layers to neural networks has in no small part contributed to the deep learning revolution in machine learning. The most successful of these techniques in the image classification domain is the *batch normalization* (BatchNorm) layer (Ioffe & Szegedy, 2015), which works by normalizing the univariate first and second order statistics between layers.

Batchnorm has seen near universal adoption in image classification tasks due to its surprisingly multifaceted benefits. Compared to an unnormalized network, its has been widely observed that using batch norm empirically results in:

- Stability over a wide range of step sizes
- Faster convergence (particularly with larger step sizes)
- Improved generalization

The multiple effects of BatchNorm make it both hard to replace and hard to analyze. In this paper we introduce Equilibrium Normalization (EquiNorm), a normalization that works in weight space and still uses a form of batch statistics unlike previous weight space approaches. EquiNorm results in very rapid convergence, even more so than BatchNorm, however as we will show in our experiments, this also results in a tendency to overfit. When combined with additional regularisation, EquiNorm can significantly outperform BatchNorm, which benefits less from this additional regularisation.

## 2 Related Work

A number of normalization layers have been proposed that can be considered alternatives to batch normalization. Batch normalization has also been extended as batch renormalization (Ioffe, 2017) to handle smaller batch sizes.

**Layer/Instance Normalization** A simple modification of BatchNorm involves computing the statistics independently for each instance, so that no averaging is done across each mini-batch, instead averaging either across channels (layer norm) or separately for each channel (instance norm). Unfortunately these techniques are known to not generalize as well as batch norm for convolutional neural networks (Sec 6.7; Sec 4.1. Jimmy Lei Ba, 2016; Yuxin Wu, 2018).

**Group Normalization** A middle ground between layer and instance normalization can be found by averaging statistics over small groups of channels. This has been shown empirically to be superior to either approach, although there is still a gap in generalization performance (Yuxin Wu, 2018). Like the approaches above, it avoids a dependence on batch statistics, allowing for potentially much smaller batches to be used without a degradation in generalization performance.

**Weight Normalization**   Additional stability can be introduced into NN training by constraining the norm of the weights corresponding to each output channel/neuron to be one. When this is done by an explicit division operation in the forward pass, rather than via an optimization constraint, this is known as weight normalization (Salimans & Kingma, 2016). An additional learnable scaling factor is also introduced. Unfortunately, to match the generalization performance of BatchNorm on image classification tasks such as CIFAR-10, this technique needs to be used together with partial (additive only) BatchNorm (Section 5.1, Salimans & Kingma, 2016).

**Local Response Normalization**   A precursor to batch norm, local normalization methods (Jarrett et al., 2009; Lyu & Simoncelli, 2008) played an important part in the seminal AlexNet architecture (Krizhevsky et al., 2012), and were widely used before batch norm was introduced. LR normalization has similarities to group norm in that it uses a group of neighboring channels (with ordering set arbitrary at initialization) for normalization. Although it aids generalization in a similar manner to BatchNorm, it does not accelerate convergence or allow for larger step sizes to be used (Sec 4.2.1, Ioffe & Szegedy, 2015).

## 3   ASSUMPTIONS

The EquiNorm method functions by modifying the weights of a convolution before it is applied. For justifying the form of our method, we make the following assumptions about this convolution, which we will discuss relaxing after detailing the method:

1. All inputs to the convolutional layer are positive, such as when the layer is preceded by a ReLU.
2. The convolution has stride one.
3. Cyclic padding is used.
4. All weights are non-zero, and there exists at least one positive and one negative weight per output channel.

## 4   METHOD

Consider initially for simplicity a convolutional layer with a single input and output channel. Let

$$w : \texttt{kernelheight} \times \texttt{kernelwidth},$$

be the weight kernel for this neuron, and let

$$x : \texttt{batchsize} \times \texttt{height} \times \texttt{width},$$

be the input tensor for a single mini-batch. We will compute scalar quantities $s$ and $b$ that modify the weights as follows:

$$w'' = sw' = s\left(w + b\right).$$

This transformation will be fully differentiated through during the backwards pass (using automatic differentiation) so that the gradient of $w$ is correct. As with BatchNorm, we also include an additional affine transformation after the convolution to ensure no expressivity is lost due to the normalization operation.

The core idea of equilibrium normalization is to balance the contribution of positive and negative weights to the output of the convolution. To this end, we introduce additional notation to address the positive and negative weights separately. Let superscripts $+/-$ (i.e. $w^+/w^-$) indicate sums of the positive/negative elements respectively. Also, let $v$ be the sum of the input data to the layer,

$$v = \sum_{i,j,k} x_{ijk}.$$

As we have two constants to determine, we need two constraints that we wish to be satisfied. The first constraint we introduce is common with batch normalization, a constraint on the mean of the

output. Since under the cyclic padding assumption, each weight is multiplied by each input element, we can constrain the mean of the output to be zero by requiring that:

$$vs \sum_{j,j} (w_{jk} + b) = 0,$$

$$\therefore b = -\text{mean}(w).$$

The second constraint controls the magnitude within the total output of the layer, of the positive weight elements:

$$vsw'^+ = r,$$

$$\therefore s = \frac{r}{vw'_+}.$$

The constant $r$ is set so that the contribution is of average 1 per output element, which is achieved by setting $r$ to the product of batch-size, output width and output height. Note that due to the mean constraint, this automatically results in the negative weight contribution also being of magnitude $r$.

## 4.1 FULL CASE

When multiple input channels are used, each weight no longer multiples each input, rather they each multiply only inputs from a single channel. To compensate for this we need to compute per-channel sums $v_c$ (where $c$ is the channel index) and change the second constraint as follows:

$$\sum_{c}^{\text{channelsin}} v_c s w_c'^+ = r.$$

The first constraint changes in the same fashion.

When multiple output channels are used, we just duplicate this procedure applying it to each channel's weights separately. We thus maintain a $s$ and $b$ value per output channel, and compute as intermediate values a $w'^+$ of matrix shape. For completeness we give the full equations below. All summations are over the full range of the summed indexes.

**Tensor shapes**

$$w : \text{channelsout} \times \text{channelsin} \times \text{kernelheight} \times \text{kernelwidth}$$
$$x : \text{batchsize} \times \text{channelsin} \times \text{heightin} \times \text{widthin}$$
$$w'^+ : \text{channelsout} \times \text{channelsin}$$
$$v : \text{channelsin}$$
$$s, b : \text{channelsout}$$

**Updates:**

$$v_c = \sum_{i,j,k} x_{icjk},$$

$$w_{dc} = \sum_{j,k} w_{dcjk},$$

$$r = \text{batchsize} \times \text{heightout} \times \text{widthout} \times \text{stride}^2$$

$$b_d = -\frac{\sum_c v_c w_{dc}}{(\text{kernelheight} \times \text{kernelwidth}) \sum_c v_c},$$

$$w_{dc}'^+ = \sum_{j,k} (w_{dcjk} + b_d) I[w_{dcjk} + b_d > 0],$$

$$s_d = \frac{r}{\sum_c v_c w_{dc}'^+},$$

$$w''_{dcjk} = s_d (w_{dcjk} + b_d).$$

At test time, we follow the technique used in BatchNorm of using a running estimate of the data statistics ($v_c$ in our method) that is computed during training time.

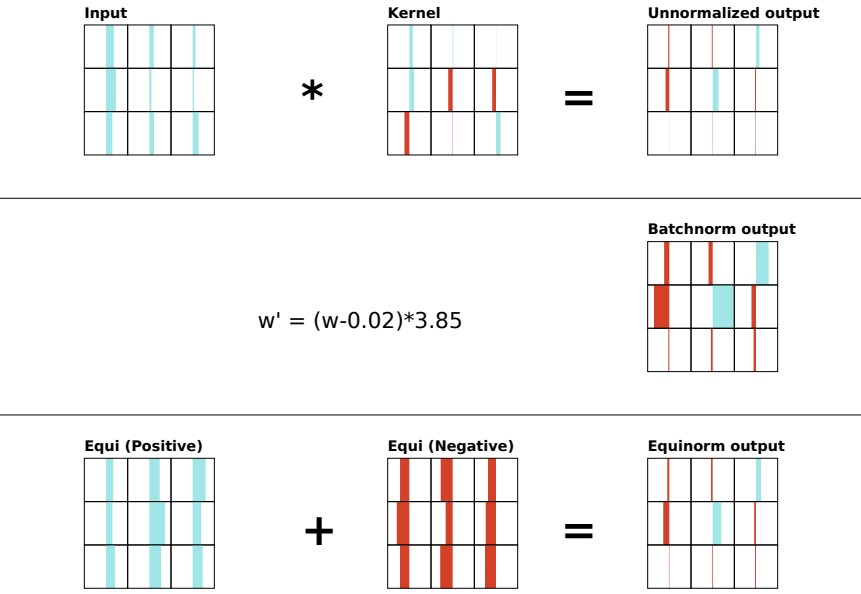

Figure 1: Equilibrium Normalization ensures the contribution from positive and negative kernel weights to the output remains of the same total magnitude, both compared to each other and between epochs. In the case shown of a $3 \times 3$ kernel (cyclic convolution) against a $3 \times 3$ image with padding 1, this magnitude is width$_\text{out} \cdot$ height$_\text{out} = 9.0$.

### SINGLE PASS FORMULATION

The above calculation requires two passes over the weights, first to compute $b$, then to compute the sum of positive elements after the addition of $b$. We can do an approximate computation using only one pass by assuming the sign of each element does not change after the addition of $b$. The $s$ calculation changes as follows:

$$
\begin{aligned}
n_{dc}^+ &= \sum_{j,k} I[w_{dcjk} > 0], \\
s_d &= \frac{r}{\sum_c v_c \left( w_{dc}^+ + b_d n_{dc}^+ \right)}.
\end{aligned}
$$

We use this variant in all experiments that follow.

## 5 DISCUSSION

### 5.1 CONTROLLING COVARIATE SHIFT

The original justification for batch normalization is its ability to minimize covariate shift, although there is some debate on whether or not this is the main contributing factor to its effectiveness, see Santurkar et al. (2018). In this context, covariate shift refers to the change between steps of the statistics of the outputs of a layer.

Like batch normalization, the approach we propose also controls the shift in the outputs of a layer between steps, just in a different way. Our approach is motivated by a hypothesis that it is not necessary to control the mean and variance precisely; other notions of scale and shift may work as well or better. Santurkar et al. (2018) show that normalizing by other norms, such as $L_1$, can work well, supporting this hypothesis.

The sum of the output from positive and negative weights is a form of $L_1$ control which can be contrasted with the $L_2$ control that Batchnorm uses. This control can be motivated by Young's convolution inequality, which bounds the output of a convolution operation in terms of the norm of the input:

$$\|x * w\|_r \leq \|w\|_p \|x\|_q,$$

$$\text{where } \frac{1}{p} + \frac{1}{q} = \frac{1}{r} + 1.$$

Note that $p, q, r \geq 1$ is also required, and that this only applies directly when there is a single input and output channel, which we assume in the remainder of this section for simplicity.

For EquiNorm, we have assumed that the input is positive, so that our input sum is equivalent to the $L_1$ norm of the input. Additionally, after subtracting off the mean, the weight vector $w'$ has $L_1$ norm equal to $w'^+ - w'^- = 2w'^+$, so we are also normalizing the weights by the $L_1$ norm. In effect, we are applying Young's convolution inequality with $p = q = r = 1$.

It is also possible to apply the above inequality with $p = 2$, $q = 1$ and $r = 2$. I.e. normalize the weights using the $L_2$ norm, giving a bound on the $L_2$ norm of the output in terms of the $L_1$ norm of the input. This is less satisfying as one convolution's output is the input of another convolution (after passing through scaling & a nonlinearity) so we would like to use the same norm for both inputs and outputs. The related weight normalization (WN, Salimans & Kingma, 2016) method normalizes weights by their $L_2$ norm, and differs from our method by centering outputs using an additional mean-only output batchnorm. Additionally, since it doesn't normalize by the input norm, the output norm can be correspondingly large. These differences have a significant effect in practice.

## 5.2 Assumptions

### Input to the convolutional layer is positive

This assumption is not necessary for the implementation of our method, rather it ensures that the output is more constrained than it otherwise would be. When ReLU nonlinearities are used in the standard fashion, this assumption holds except in the first layer of the network where input pixels are usually in the range [-1,1], due to pre-normalization. We recommend this pre-normalization is removed, as it is unnecessary when normalization happens immediately inside the first convolution. Recommendations in the literature that suggest input normalization is beneficial are usually referring to networks without per-layer normalization.

### Non-strided convolutions

A strided convolution can thought of as a non-strided convolution with the extra output values thrown away. If Equilibrium normalization is used with a strided convolution, the contribution to the output from positive and negative weights will no longer be exactly balanced. In practice the violation will be small if the output of the non-strided version of the convolution is smooth.

### Cyclic padding

Our equations for $b$ and $s$ assume that each weight for an input channel is multiplied by each input value for that channel. Most deep learning frameworks use zero-padded convolutions instead of cyclic padding, which violates this assumption. In practice we do not find this violation to be troublesome, as it only affects edge pixels, and has a dampening effect as it only reduces the output contribution of the positive or negative weights.

### All weights are non-zero, and there exists at least one positive and one negative weight per output channel

We avoid the use of an $\epsilon$ parameter such as used in BatchNorm, as the denominator of our normalization factor is only zero if *every* weight for every input channel is simultaneously positive (or all negative), or the weights become extremely small. The later case does not appear to happen in practice. Nevertheless, we find it helps to initialize the weights in a balanced fashion, so that no channel's

weight kernel is all positive or all negative. We do this by modifying the default initialization by resampling any such kernel-weight's signs.

# 6 EXPERIMENTS

In our plots we show a comparison to BatchNorm and GroupNorm. We omit a comparison to Layer/Instance Normalization as our initial experiments were consistant with findings in the literature that show that they are inferior to GroupNorm and BatchNorm, at least for the convolutional architectures we consider below (Yuxin Wu, 2018). We also performed a comparison against the WeightNorm method, however despite significant efforts we were not able to get it to reliably converge when using very deep architectures such as ResNet-152 that we choose for our experiments. We could not find any results in the literature where it is sucessfully applied to state-of-the-art deep networks, and we believe this is a real limitation of the method.

## 6.1 CIFAR-10/100

The CIFAR-10 dataset (Krizhevsky, 2009) is considered a standard benchmark among image classification tasks due to its non-trivial complexity, requiring tens of millions of weights to achieve state-of-the-art performance, and also its tractable training times due to its small size (60,000 instances). The downside of this small size is that significant care must be taken to avoid overfitting. This overfitting can be partially avoided using data augmentation, and we followed standard practice of using random horizontal flips and crops (pad 4px and crop to 32px) at training time only.

Our initial experiments involving a non-bottleneck wide ResNet network with 20 convolutions, 96 initial planes after first convolution and 9.7m parameters. The hyper-parameters used were LR: 0.1, SGD+Mom: 0.9, decay: 0.0001, batch-size: 128, 1 GPU, 10 fold learning reductions at epochs 150 and 225, and standard fan_out normal initialization following He et al. (2015). These parameters are defaults commonly used with BatchNorm and were not tuned.

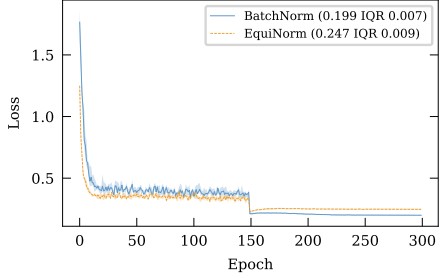

Figure 2: Indications of overfitting, as test loss starts to increase significantly after the first learning rate decrease.

Our experiments indicated that our EquiNorm approach converged significantly faster than Batch-Norm, but also overfit significantly more (Figure 2).

We believe this is caused by the batch statistics having less noise with EquiNorm than BatchNorm, rather than the faster initial convergence, as experiments involving reduced step sizes did not further improve generalization. Similarly, we were not able to achieve comparable fast initial convergence by using larger step-sizes with BatchNorm.

We found instead that we could match the generalization of BatchNorm using either of the following two approaches:

1. Using fewer instances to compute the batch statistics. Using the first quarter of the batch to compute the statistics used for the full batch successfully fixed the overfitting seen in Figure 2, resulting in higher test accuracy for EquiNorm (95.8%) over BatchNorm (94.7%) and no overfitting visible in test loss.

2. Using mixup (Zhang et al., 2018) or manifold mixup (Verma et al., 2018), which introduce activation noise of a similar nature.

We recommend the use of manifold mixup, as it significantly improves test accuracy for both Batch-Norm and EquiNorm, although it does result in higher test loss in some cases.

Following closely the approach of Verma et al. 2018, we applied both EquiNorm and BatchNorm to the larger near state-of-the-art pre-activation ResNet-152 architecture (He et al. 2016a, 58.1m parameters, [3,8,36,3] bottleneck blocks per layer respectively, 64 initial channels), using a modified

version of their published code and the hyper-parameters listed above, with manifold mixup used for each method. As Figure 4a shows, the test set performance is essentially the same at the final epoch, but EquiNorm converges significantly faster at the early epochs.

CIFAR100

We also achieved a similar performance on the CIFAR-100 dataset (which has similar properties to CIFAR10) as shown in Figure 4b, where we used the same hyper-parameters and network architecture as for CIFAR10.

## 6.2 SHORTER DURATION TRAINING

Given the encouraging results above during the early stages of optimization, we investigated if EquiNorm was superior when training is restricted to 30 epochs instead of 300. We used a "super convergence" learning rate schedule as suggested by Smith & Topin (2017), consisting of a 5 fold ramp in learning rate (starting at 0.1) from epochs 1 to 13, then a 5 fold ramp down to epoch 26, followed by further annealing by 100x down over the remaining epochs. Momentum follows a reverse pattern, from 0.95 to 0.85 to 0.95, and fixed at 0.85 after epoch 26. Manifold mixup was used again for both methods. Using this schedule EquiNorm shows a 94.0% (IQR 0.22) median test accuracy compared to 93.3% for Batch-Norm (IQR 0.77).

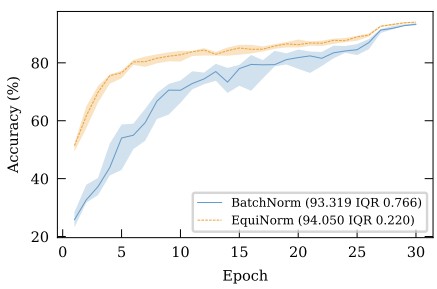

Figure 3: Shorter duration CIFAR10 training (WRN network)

## 6.3 STREET VIEW HOUSE NUMBERS

The SVHN+EXTRA dataset (Netzer et al., 2011) is much larger than CIFAR-10/100 (73,257 + 531,131 training instances), so we trained across 2 GPUs, using $2\times$ larger mini-batches (size 256) so as to keep the batch statistics noise (which are computed on a per-gpu basis) the same. Other hyper-parameters were also kept the same, with the exception that we trained for fewer epochs (with LR reductions moved to epochs 80 and 120). Figure 4c shows that EquiNorm achieves essentially the same generalization performance as BatchNorm. On this problem GroupNorm appears inferior, although this may be due to the default group-size of 32 being suboptimal here.

## 6.4 ILSVRC 2012 IMAGENET

We also ran some preliminary experiments on the ILSVRC 2012 ImageNet classification task using the standard ResNet50 architecture (He et al., 2016b). Our results here show a generalization gap between EquiNorm and BatchNorm/GroupNorm. It may be possible to eliminate this gap using additional regularisation as in the CIFAR-10 case, however we found manifold mixup to not yield such an improvement.

## 6.5 REPORTING TRAINING VARIABILITY

We are careful to report results aggregated over enough runs involving different RNG seeds so that run-to-run variability does not effect our conclusions. This is absolutely necessary for the smaller test problems above as the differences between runs can be comparable to the difference between the compared normalization methods, and indeed differences in reported results in the literature. We report median and inter-quartile statistics (i.e. our plots include point-wise 25% and 75% percentile ranges of the values seen), as these are more representative of actual performance, and particularly the asymmetry of test accuracy variability. The more commonly used two-standard-deviation bars based on a normal assumption can show values both below and above actually seen data (such as $> 100\%$ accuracy upper bounds), and are not supported by statistical theory for small samples such as the ten used here.

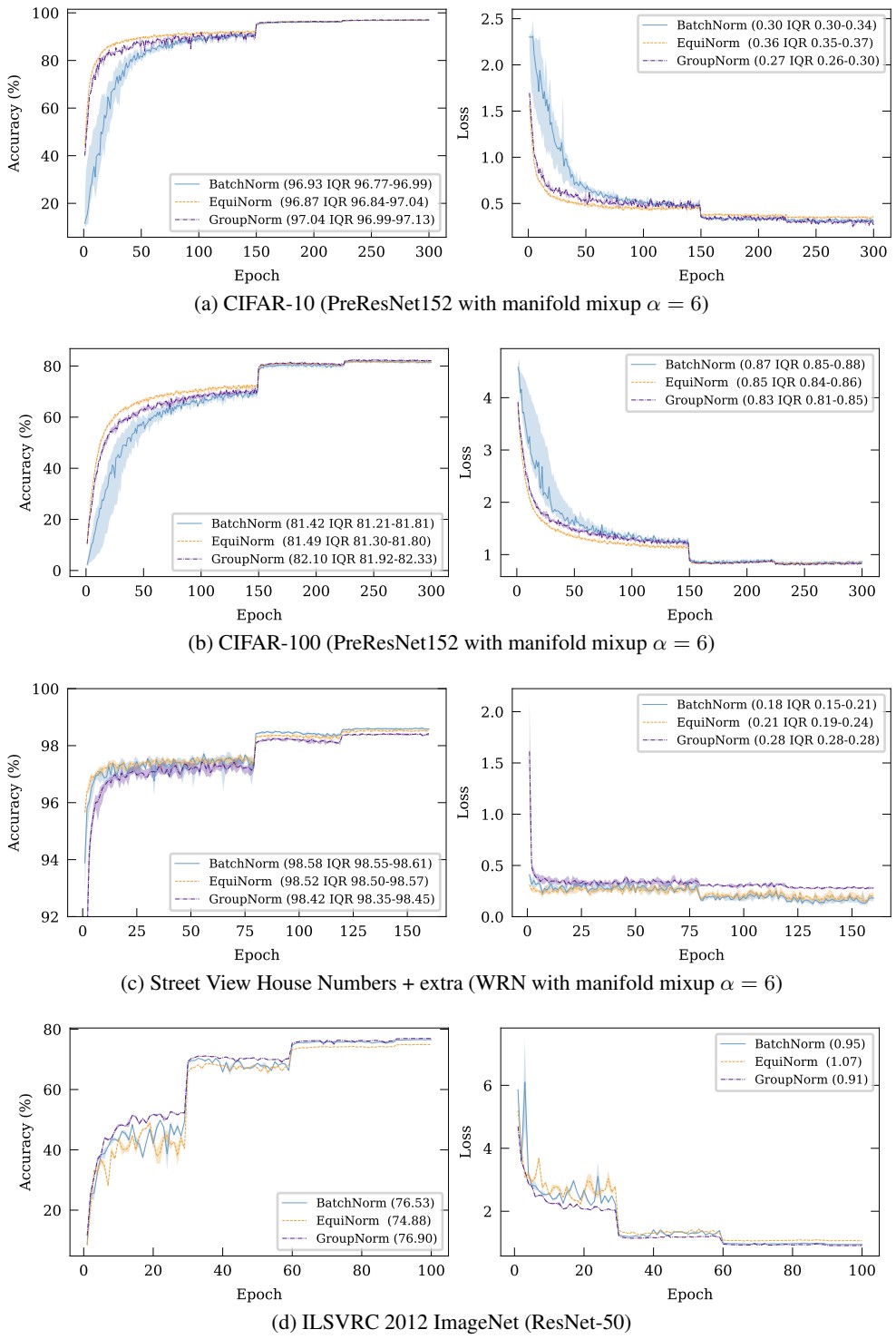

(a) CIFAR-10 (PreResNet152 with manifold mixup $\alpha = 6$)

(b) CIFAR-100 (PreResNet152 with manifold mixup $\alpha = 6$)

(c) Street View House Numbers + extra (WRN with manifold mixup $\alpha = 6$)

(d) ILSVRC 2012 ImageNet (ResNet-50)

Figure 4: Test set accuracy and loss. Median of 10 runs shown with interquartile regions overlaid with the exception of the ImageNet plot which uses 3 runs.

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
