# OpenReview forum: "CONTROLLING COVARIATE SHIFT USING EQUILIBRIUM NORMALIZATION OF WEIGHTS"
_ICLR.cc/2019/Conference_

### Official Review · AnonReviewer3 · 2018-10-31
**Elegant idea but experiments not fully convincing**

**Rating:** 7
**Confidence:** 4

**Review:**

This manuscript introduces a new layer-wise transform, EquiNorm, to improve upon batch normalization. As with batch normalization and related techniques, the idea is to introduce a simple linear transform at each layer to reduce the dependency of the features to the data. Unlike batch normalization, the procedure does not modify the inputs to the layers but rather the layer weights. For this purpose, a scaling factor and a shift is computed on a mini batch, separating positive and negative weights to compute easily both running estimates of shift and of spread (here in the l1 sense). The method is compared to BatchNorm and GroupNorm on several classic computer vision datasets. Empirically, the method converges faster in the beginning of the optimization, in the sense that in the first few epochs the test accuracy is higher than for BatchNorm. However, this benefit decreases with more epochs and when the results are close to peak performance the difference between methods is a small. In addition, test accuracy can decrease at the end of the optimization, which the authors interpret as a sign of increased overfit, and tackle with clever data augmentation. The paper reads well.

The strengths of the paper are the elegance of the solution proposed, and the faster convergence. To me, the main drawback of the manuscript is that the results do not show a clear benefit at convergence of EquiNorm compared to BatchNorm. It seems that as the learning rate decreases there is no consistent difference: in none of the 4 datasets does EquiNorm give a gain larger than a quartile of the distribution of scores and in 2 out of 4 it performs less well.

My advice to improve this work would be to do more experiments and to show better that the lack of performance gain is due to overfit, and can be fixed with larger data or data augmentation.


The faster initial convergence could be a real benefit of this method. However, the time to useful convergence is the same than with BatchNorm. This is probably because the learning-rate decrease schedule is the same. It might be interesting to study adapted learning-rate decrease.

To argue for increased overfit, it would have been interesting to compare test error with train error, and to plot train loss alongside with figure 2.

While the fast convergence is a benefit, it would have been interesting to also compare EquiNorm with BatchNorm with learning scale schedules (ie outside "super convergence" settings). While costly, such an experiment would help separating the benefits of EquiNorm from choice of learning rate and give a standard baseline to compare to. Ideally, EquiNorm achieves this baseline with less compute time.

I would like to congratulate the authors for reporting multiple runs with different RNGs and the quantile of the distribution. This is absolutely good practice to judge the significance of improvements and is seldom done.

Why does figure 2 shows only test loss, while the other experimental figures show test accuracy and loss?

---

> ### Author Response · Authors · 2018-11-14
> **Rebuttal**
>
> Thank you for the insightful comments. We agree that a larger and more systematic exploration of the combined effects of learning rate schedules and normalization would be interesting. We will look into this in future experiments.
>
> You are correct that we only show a benefit to using EN other BN in the heavily time-constrained regime, where only a small number of epochs are possible. The EN method is still useful outside this setting, as we believe the way it separates the optimization effects of BN from it's regularization effects is of independent interest. The fact that this separation is possible, and that each of these two components may be individually improved is one of our contributions.

---

### Official Review · AnonReviewer1 · 2018-11-02
**logical new approach to covariate shift problem**

**Rating:** 6
**Confidence:** 1

**Review:**

Authors present a new normalization technique called Equi-Norm and experimentally show its fast convergence properties over its popular competitors Batch-norm and Group-norm. The main idea in the paper is the EquiNorm method modifies the weights of a layer before forward propagating through it such that the contribution from positive and negative kernel weights to the output is same. In the experimental section, their approach is shown to be more accurate than Batch-norm and group norm on all datasets except, Imagenet. Is there any reasoning why this method performed slightly poorly on this particular dataset.

Quality:
Results are convincing and comprehensive; authors compare their proposed approach to major two baseline normalization frameworks and demonstrate improved performance.

Clarity:
The paper is well written for readers to understand however there are many grammar mistakes in multiple places.

Originality:
The work seems to be a logical new approach to covariate shift problem and as described above, I think the proposed method provides convincing results.

I am personally not as familiar with normalization approaches so my confidence in the assessment is low; my main experience is in the computer vision for autonomous driving and sparse coding.

---

### Official Review · AnonReviewer2 · 2018-11-02

**Rating:** 4
**Confidence:** 4

**Review:**

This paper introduces a normalization technique, which normalizes the weights of convolutional layers. The method looks like the combination of batch normalization(BN) and weight normalization. Several assumptions are provided to proceed what the authors would like to show, and some experimental results follow.

pros)
(+)  The idea of normalizing weights looks good.

cons)
(-) The major problem is the experimental section, where the results are not convincing and cannot support the effectiveness of the proposed method. Specifically, the proposed method cannot outperform batch normalization (BN) in respect of validation/test accuracy even the authors claim that training is faster. It looks similar to the training curve when using smaller learning rates.
(-) The proposed method (+ mixup or manifold mixup) cannot show better results over batch normalization on ImageNet dataset.
(-) The proposed method seems to have a dependency on the additional method such as mixup and manifold mixup. Furthermore, the parameter of them could determine the overall performance of it.
(-) Section 5.1 cannot fully support the conjecture which is that the proposed normalization method can concern the covariate shift effect.

comments)
- What is the benefit of the faster convergence in early epochs? We can observe faster convergence when using smaller learning rates, but finally, the larger learning rates can show better validation/test accuracy. As shown in Figure 2,  and some results in Figure 4, the accuracy does now outperform BN.  In this light, the faster convergence does not seem to have any advantages.
- The authors should put the training curve about the first approach (shown on page 6) to clarify how the training goes differently.
- For the second approach on page 6, it looks interesting the proposed method can outperform BN on CIFAR datasets when combining with mixup or manifold mixup. However, there is no analysis of why it works.
- The first equation would be changed without the cyclic padding assumption, and this would cause errors normalizing all weights (just following the derived equations). So, this affects not only on edge pixels but all the pixels in features.
- Updating r with "batchsize x heightout x widthout x stide^2" makes a tremendously big value in the earlier layers (i.e., close to the input), so it must affect the normalization results.


The paper contains an interesting approach by normalizing weights directly, but the grounds for the conjecture are not clearly addressed. A few minor things for better readability should be addressed: all the equation should have equation number so that the reader can easily follow the paper, and the subsection titles (e.g., 4.1 Full case, 4.2 Assumptions (for what?)) are quite hard to understand what the authors will tell us.

---

> ### Author Response · Authors · 2018-11-14
> **Rebuttal**
>
>
> Thanks for the detailed comments. We would like to respond to a few points, with the hope that you will reconsider your evaluation.
>
> • In the cons section you say that “It looks similar to the training curve when using smaller learning rates.”. As we state in the results section, no value of the learning rate for BN/GN gives as fast a convergence rate as we see for EN. We show the curves corresponding to the best rate. Almost any slower method will have a curve looking like the result of using a smaller learning rate, as there are few other shapes it can take.
>
> • We do not claim to get better validation accuracy. Given our method introduces no new parameters into the model, and is not applying a regularization effect, it can't be expected to lead to higher accuracy. Our goal is optimization speed, which should be worthy enough. I hope you reconsider the significance of our contribution in this regard.
>
> • The poor results on imagenet are definitely a down-side of our method. The fact that we tested on imagenet and provided the negative results should be taken into consideration, as many ICLR papers still only test on MNIST/CIFAR10, and often negative results are omitted entirely.
>
> • The requirement of mixup should not be seen as a down-side. By separating the regularization effect of batchnorm from it's optimization effect, we provide a deeper understanding of batchnorm, and show that it is possible to improve the optimization side independently.

---

### Meta-Review · Area_Chair1 · 2018-12-16
**Well-written paper but the empirical result seem to be not fully convincing**

**Confidence:** 4
**Recommendation:** Reject

**Metareview:**

This paper introduces a technique called EquiNorm, which normalizes the weights of convolutional layers in order to control covariate shift. The paper is well-written and the reviewers agree that the solution idea is elegant. However, the reviewers also agree that the experiments presented in the work were insufficient to prove the method's superiority. Reviewer 2 also expressed concerns about the poor results on ImageNet, which calls into question the significance of the proposed method.